# Influence of Different Fiber Dosages on the Behaviour of Façade Anchors in High-Performance Concrete

**Szymon Grzesiak, Matthias Pahn \*, Milan Schultz-Cornelius and Nora Susanne Bies**

Department of Civil Engineering, Technical University of Kaiserslautern, Paul-Ehrlich-Straße 14, 67663 Kaiserslautern, Germany; szymon.grzesiak@bauing.uni-kl.de (S.G.); milan.schultz-cornelius@bauing.uni-kl.de (M.S.-C.); nora.bies@bauing.uni-kl.de (N.S.B.)
\* Correspondence: matthias.pahn@bauing.uni-kl.de

**Abstract:** The behaviour of façade anchors in high performance fiber reinforced concrete (HPFRC) has not been investigated in sufficient detail in recent years. The regulations in the European Technical Approvals also do not fully describe the load-bearing capacity of anchor systems. Due to the increase in the production of HPFRC elements, it is necessary to analyse the impact of added fibers in the concrete composition on the behaviour of anchors. In particular, the behaviour of anchors in filigree façade elements, which is one of the main application areas of the programme of polypropylene (PP) fiber-reinforced concrete, is therefore analysed. With a sufficient content of PP fibers surrounding the steel anchors oriented in an optimal direction, the fibers may enhance both the load-bearing capacity of anchors and the ductility of concrete. However, unfavourable effects on the installation process or even on the load-bearing capacity may also occur due to unfavourable fiber orientation. Therefore, tensile and punching tests were carried out in uncracked concrete with different types of anchor systems containing a tension anchor and an adjustable spacer bolt. The PP fiber content of the concrete component varied during the tests.

**Keywords:** high performance fiber reinforced concrete (HPFRC); polypropylene fiber (PP); façade anchors

## 1. Introduction

The façade separates the interior from the exterior, ensuring a comfortable indoor climate. As an intermediate element, the façade is thus exposed to internal and external factors and serves as protection against wind, rain and heat. Therefore, the façade has an impact on the energy efficiency of the building. Furthermore, the façade is an important design element affecting the appearance of the building and can be realised in many different forms and styles [1].

Façade cladding is a rapidly growing product group for the pre-cast concrete industry. It is therefore of great interest to optimise the load-bearing capacity and economic efficiency of these façade systems [2]. As a rule, the façade panels of the façade systems should have exposed concrete properties, i.e., they should remain in uncracked condition under load. To meet this quality requirement in concrete façade panel, the HPC and UHPC were, in the recent years, the focus of research investigations [3]. Innovations and developments included the understanding of material properties and the production process of the façade elements. In the centre of attention are reduction in façade thickness and therefore the limitation of $CO_2$ emission during the cement production. The environmental benefits are also provided by reducing the amount of steel reinforcement in non-structural as well in structural applications [4]. Schultz-Cornelius [5,6] characterised UHPC as the material for concrete wall panels under different exposure and geometric influences. Unfortunately, the ductility of the material decreases with the increasing concrete strength. To improve the macroscale structural performance of concrete members, different fibers are dosed into the concrete mix [7]. The positive effect of fiber addition in concrete is fracture toughness [8],

which is visible in the form of prevention from crack formation and propagation in loaded elements [9]. The thin façade panels made of fiber concrete meet the current architectural and environmental requirements. By omitting conventional reinforcement, it is easier to create complex façade shapes and geometries based on slim elements [10]. Likewise, this system can be used to create large façade surfaces that are easily adapted to different design possibilities.

The development of standards and regulations has accelerated over the last decades due to the increased use of fiber concrete. A closer look at the standards and regulations quickly reveals that although the design models are similar, not every model is optimised for every application [11–14]. The main reason for using fibers in HPC is to increase the material ductility and reduce the brittleness [15]. Moreover, the different fiber content will allow resources to be saved in the concrete elements. The material optimisation is affected by dosages of fiber in concrete matrixes [16]. Mechanical parameters cuch as compressive, bending and tensile strength of HPFRC and the optimal behaviour of the steel anchor in uncracked condition enables the optimised design of filigree, safe and lightweight elements. Baby at al. [17] draws attention to the durability of HPC elements, which is ensured by the small size of additives in concrete. They allow better compactness of the hardened concrete.

Innovations and developments in the research of concrete façades allow for the use of new anchoring systems. More details about the requirements of anchoring systems are provided in research by Pahn [10] and by Carstens [18]. Figure 1 presents two possible anchoring systems for façade elements, which differ from each other by their connection parts. The first group contains point-supported panels and the second group contains edge-supported panels. Figure 1a presents point-supported façade panels, which are connected to the structure at no less than four points. The point-supported façade panels can be designed with considerably more anchoring points. Point-supported panels may be realised either by long rod-shaped fasteners made of glass fiber reinforced plastic or steel by undercut anchors with a combination of a substructure made of aluminium or by a combination of suspended access anchors as well as tensile and compressive spacer bolts.

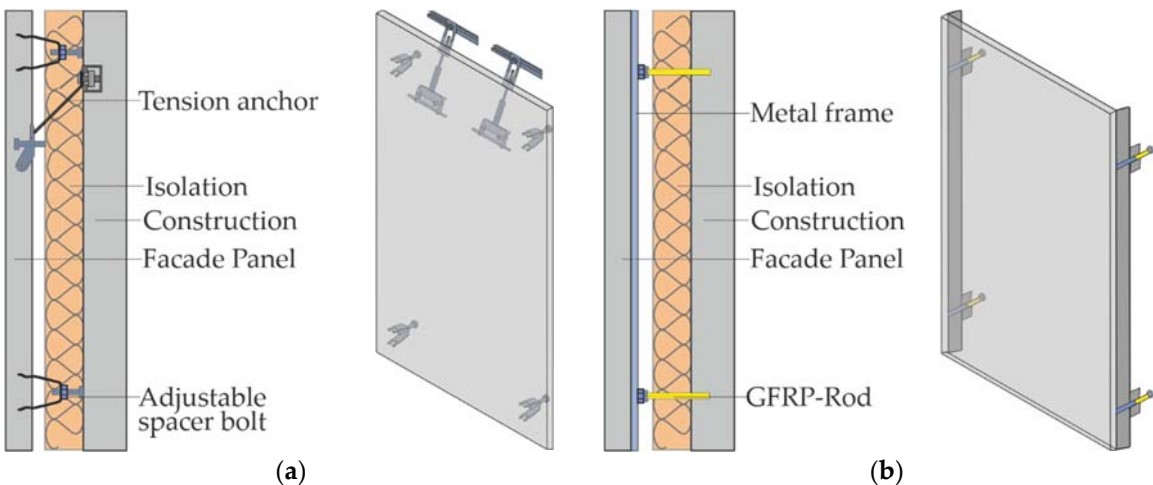

**Figure 1.** (**a**) Point-supported façade panels with tension anchor and adjustable spacer bolt [19]. (**b**) Schematic depiction of the edge support of a façade with GFRP-bars [5].

For edge-supported façade panels (detail Figure 1b), the load is transferred over the edges of the panel. In this case the façade is fixed in a metal frame. On this frame, the façade panel is connected to the existing wall at certain intervals with rod-shaped fasteners made of glass fiber reinforced polymers (GFRP) and aluminium brackets [5].

While several papers have been published in recent years in the field of anchoring in masonry structures and concrete slabs, the behaviour of façade anchors in HPFRC has

not been actively studied. Nonetheless, few research have been gathered to explore the properties of anchor system in hardened concrete. Klug et al. [20] tested undercut anchors in steel fiber reinforced concrete. Concrete was classified based on its compressive strength and assigned to a maximal strength class C50/60. Kurz et al. [21] studied different kinds of anchor systems in noncracked concrete. Tension tests were carried out using the varied content of steel fibers. In the laboratory tests, Ivorra et al. [22] examined the anchoring systems of their mechanical characterisation. In their research, the results were compared with the analytical models developed by the Finite Element Method (FEM). Steel anchors were also investigated by Poveda et al. [23]. Their research investigated the mechanical response and the failure modes by analysing crack propagation. In their paper, Dizhur et al. [24] investigated the failure mode of adhesive anchor connection in unreinforced clay brick walls. Tóth et al. [25] carried out tension and shear tests on single anchor as well on anchor groups in HPFRC. Use of disperse reinforcement results in higher concrete cone capacity. By performing numerous tests, the authors proposed a modification factor that is related to the dosage of steel fibers dispersed in the concrete. The same hooked-end-type steel fibers were used in the study of Bokor et al. [26] for the preparation of a sample in the experimental program with the behaviour of single fasteners in concrete. Based on pull-out tests under quasi-static and moderately high loading rates, the increase in concrete cone capacity in comparison to the measured concrete cone capacity in normal plain concrete was presented. Mahrenholtz et al. [27] presented similar higher ultimate loads and developed larger corresponding displacements for FRC in comparison to the normal plain concrete.

## 2. Research Significance

The aim of this work is to investigate the influence of different fiber dosage on the load-bearing behaviour of façade anchor in high-performance concrete. This study would contribute to the development of the façade system with fiber instead of traditional steel reinforcement. The brittle failure of the high-performance concrete will be reduced through the dispersed fiber reinforcement. Due to the higher fiber dosage, an increase in the diameter of the concrete cone around the anchor is envisaged. Furthermore, an increase in load-bearing capacity in façade anchor is expected.

## 3. Experimental Program

### 3.1. Overview of the Anchoring Concept

A test program was derived based on the specifications of the façade anchors (Table 1). It refers to the concrete mix with the different fiber dosage and effective anchorage depth $h_{ef}$. The concrete formulas and the fiber dosages were defined in tabular form (Tables 3 and 4). The numbering in the tables corresponds to the test number of the experimental part. In the project, the anchor system for point-supported façade panels was used (see Figure 2). It consists of the following components: tension anchor and adjustable spacer bolt. Adjustable suspended tension-anchor transfers the dead-load of façade panels to the main supporting structure. This anchor provides a comprehensive range of isolation thickness (detail d in Figure 2). Adjustable spacer bolt set the distance to the wall and to transfer horizontal loads.

Panel anchors in point-supported façade panels are used as a substructure in façade construction in combination with tensile and compressive spacer bolts. An anchoring system consists of four tensile/compressive spacer bolts (b in Figure 2) and two panel anchors (a in Figure 2). The tensile/compressive spacer bolts are used for the transfer of horizontal loads such as wind suction or wind pressure. The panel anchors, on the other hand, are used to transfer the vertical weight of the façade panel. The tensile/compressive spacer bolts (b in Figure 2) are inserted into the back of the façade panel (c in Figure 2) at previously determined positions when the panel is concreted. The steel elements are temporary fixed to the formwork so that the planned concrete cover is $h_{ef}$ = 25 mm. When

the façade panel is installed, they are attached to the existing wall (e in Figure 2) with a screw system. This ensures horizontal load transfer.

**Table 1.** Overview of tested anchors.

| Series | Anchor and Tests | $h_{ef}$ (mm) | Fiber Dosage (kg/m³) | Number of Specimens |
|--------|------------------|---------------|----------------------|---------------------|
| 1 | Tension anchor (pull-out resistance tests) | 26 | 10 | 3 |
| | | | 15 | 3 |
| | | | 25 | 3 |
| | | | 35 | 3 |
| 2 | Adjustable spacer bolt (pull-out resistance tests) | 25 | 10 | 3 |
| | | | 15 | 3 |
| | | | 25 | 3 |
| | | | 35 | 3 |
| 3 | Adjustable spacer bolt (punching resistance tests) | 25 | 10 | 3 |
| | | | 15 | 3 |
| | | | 25 | 3 |
| | | | 35 | 3 |

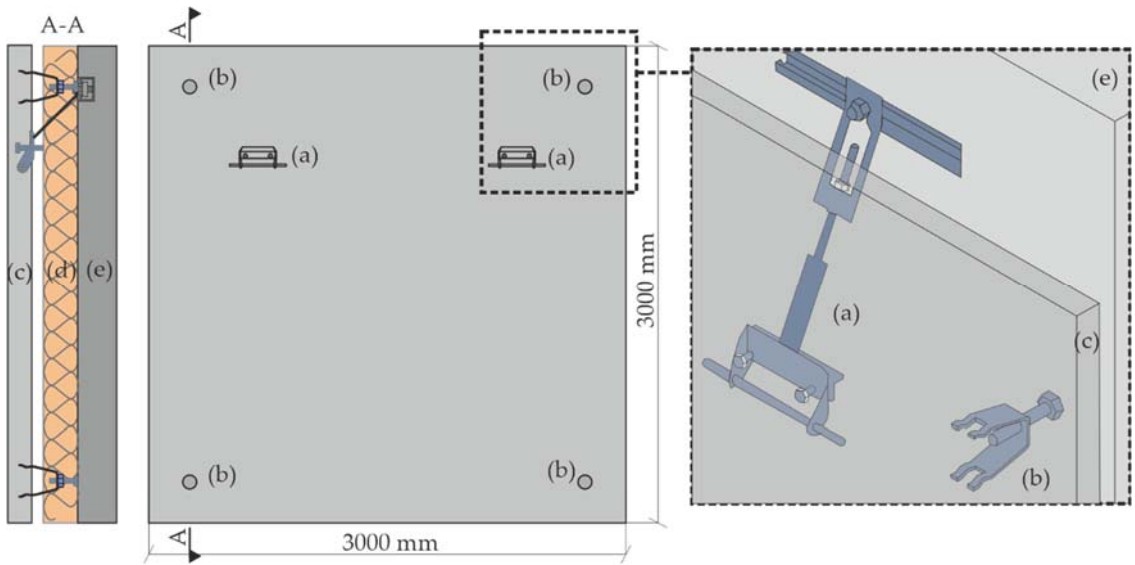

**Figure 2.** Anchoring concept with the following: (**a**) two tension anchors (for the transfer of the dead load of façade panels), (**b**) four adjustable spacer bolts (for the transfer of horizontal loads and to set the distance to the wall), (**c**) façade panel, (**d**) insulation and (**e**) wall.

To install a concrete façade panel, two façade panel anchors are required to support vertical loads (dead load) and four horizontal anchors to ensure correct wall spacing. The standard installation provides two spacer bolts at the top of the panel and two at the bottom. In the case of stacked suspended façade panels, the bottom spacer bolts can be replaced with steel dowels. Depending on the expected wind loads and the shape and size of the panel, additional suction protection may be required for horizontal anchors (for example, compression bolts and adjustable restraints).

*3.2. Materials*

3.2.1. Fiber

Among the different fiber materials such as steel, glass, carbon or basalt, polymer fiber has attracted more attention in the last years [28]. The method of fiber production significantly defines the shape of the fiber cross-section. Polypropylene fibers are produced

by catalytic chain-transfer polymerisation [29]. The material is generally characterised by its low density, easy mouldability, resistance to corrosion and low production costs. Compared to steel fibres, polymer fibres are elastic and flexible due to their chemical composition [30]. There is no possibility of injury caused by the sharpness of the steel fibers during the concreting process. Similarly, the criteria of stiffness and resistance to alkaline attack also plays a major role in different fields of application [31].

This study examines the impact of one type of fiber under variations of fiber dosage on the mechanical properties of hardened concrete: polypropylene fibers (also referred to as PP fibers). In this study, the polypropylene fiber MasterFiber® 235 (PP) is used (see Figure 3). The characteristics of PP fibers are shown in Table 2. Compared to steel, non-corrosive PP fibers do not require any concrete cover in order to be protected against environmental attacks.

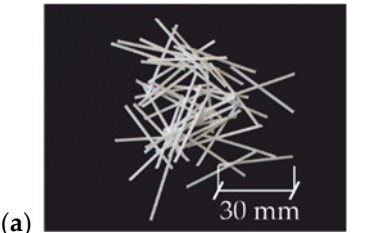  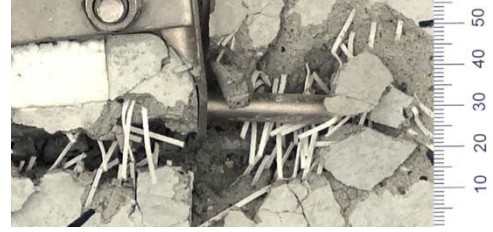

(**a**)  (**b**)

**Figure 3.** Polypropylene fiber MasterFiber® 235 (PP) used in the present study: (**a**) Fiber before concreting; (**b**) fiber in hardened concrete with dosage 35 (kg/m³) after the experimental test.

**Table 2.** Mechanical parameters of fibers used as specified by the manufacturer [32].

| Type of Fiber | Tensile Strength (MPa) | Modulus of Elasticity (MPa) | Diameter (mm) | Length (mm) | Specific Gravity (kg/m³) |
|---|---|---|---|---|---|
| MasterFiber® 235 (PP) | 500 | >8000 | 0.70 | 30 | 910 |

Polypropylene fibers are produced by polymerisation into linear macromolecules of propane. The production first succeeded in 1954 and, in 1957, production was industrialised. Depending on the production process used, the material properties may vary. Due to their low density $\rho_{PP}$ = 895 kg/m³ to $\rho_{PP}$ = 920 kg/m³, PP fibers are very light in weight. As a rule, the tensile strength of PP fibers is between 450 N/mm² and 700 N/mm² with a modulus of elasticity between 4000 N/mm² and 9000 N/mm² [33]. In order to increase the tensile strength of the fibers, they are stretched during production. PP fibers are also highly resistant to fatigue, acids and alkalis [34,35]. They are mainly used in pre-cast concrete elements and ready-mix concrete [36,37].

### 3.2.2. Concrete Mix Design

For this experimental study, four different concrete mixtures with the same amount of cement, aggregates and additives were prepared. They only differed in fiber dosage (see Table 3). Mix ID 1 contained 10 kg/m³ of MasterFiber® 235 (PP). To investigate the influence of the fiber content, Mix ID 2, 3 and 4 contain 15 kg/m³, 25 kg/m³ and 35 kg/m³ fibers each. The mixtures were prepared using a 55 L capacity horizontal forcing type concrete mixer.

**Table 3.** Test programme mixtures of concrete composition.

| Mix ID | Type of Fiber | Weight (kg/m³) |
|:---:|:---:|:---:|
| 1 | MasterFiber® 235 (PP) | 10 |
| 2 | MasterFiber® 235 (PP) | 15 |
| 3 | MasterFiber® 235 (PP) | 25 |
| 4 | MasterFiber® 235 (PP) | 35 |

The appropriate HPFRC was produced with cement CEM I 42.5 R, quartz sand, limestone powder and silica fume. Limestone powder and silica fume were used as fillers. The aggregates were dried sand 0/2 mm, quartz and basalt 1/3 mm. The grain size curve of aggregates according to [38] is presented in Figure 4b. To eliminate the segregation and bleeding risk, concrete with low consistency was used [39]. A plasticiser MasterGlenium ACE 430 was used to enhance workability. The cement content and water–cement ratio in the concrete composition were 27.449% and 0.323, respectively. The slump flow measure has a maximum value of 650 mm for dosage 35 kg/m³. The materials used in the concrete mix and their density are shown in Table 4.

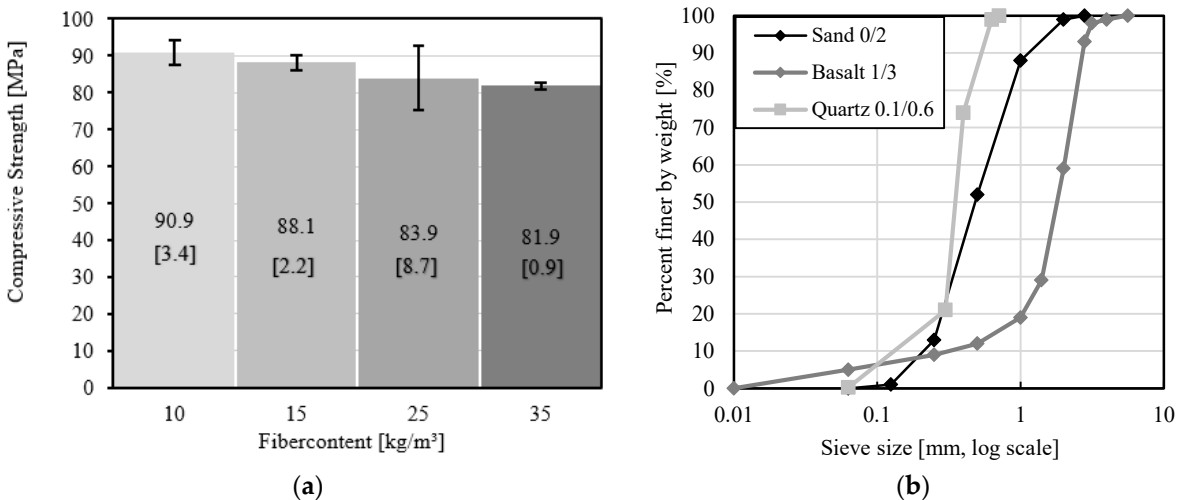

**(a)**             **(b)**

**Figure 4.** (**a**) Compressive strength of concrete (st. dev) with different fiber dosages MasterFiber® 235 (PP) at 33 days (10 and 35 kg/m³) and at 32 days (15 and 25 kg/m³) of a cube specimen 100 × 100 × 100 mm³. (**b**) The grain size curve of aggregates according to [38].

**Table 4.** The materials used in the concrete mix and their density.

| Material | Raw Density (kg/m³) | Weight (kg/m³) |
|:---:|:---:|:---:|
| Cement (CEM I 42.5 R) | 3100 | 650 |
| Aggregate 0 to 3 mm | 2600 | 990 |
| Silica fume | 700 | 50 |
| Limestone powder | 2700 | 415 |
| Plasticiser MasterGlenium ACE 430 | 1060 | 18 |
| Water | 1000 | 210 |

Compressive strength tests on hardened concrete were performed with 100 mm cubes according to EN 12390-1 [40]. The specimens were cured indoors and, on the next day, were taken out of moulds and wrapped in foil. They are stored at a temperature of 21 ± 3 °C and relative air humidity of 60%. The procedure of casting and storing did not differ between each concrete mixture. The test was carried out based on three specimens of each concrete mix. The compressive strength of the Mix ID 1 with the lowest fiber dosage at 33 days of

curing was 90.9 MPa compared to 81.9 MPa after 33 days of Mix ID 4 with the highest fiber dosage (see the compressive strength values with their coefficient of variation in Figure 4a). Damage was noticed on the edges of one cube for dosage 25 kg/m³, which explains the larger scatter for this mix. The results show that the compressive strength decreases with increasing fiber content. For the normal strength concretes, the higher dosage will increase compressive strength (research by Aslani and Meesala [41,42] with the strength class under 30 MPa), but the opposite trend is valid for higher strength concretes (research by Zhang et al. [43] with a compressive strength up to 100 MPa). The space between concrete and fibers is filled with air. Furthermore, the higher fiber dosage reduces the concrete-part in mix. A small air volume and lower proportion of the concrete determines the compressive strength and density of high-performance concrete. The reduction in compressive strength was obtained in the study by Richardson [44] on fiber reinforced concrete with a low, medium and high compressive strength.

### 3.3. Testing Procedure

3.3.1. Determination of Pull-Out Resistance

The pull-out resistance was determined on the basis of the experimental setup of a shear test presented in the "Guideline for European Technical Approval of Metal Anchors for Use in Concrete" [45]. This guideline has been prepared by the "European Organisation for Technical Assessment" and defines the test method for the European Assessment Document (EAD). Differences between the ETAG and presented tests are in specimen and anchor dimensions. Moreover, the fiber reinforcement and use of high-performance concrete are not specified in the guideline. The anchorage system requires a combined tensile and shear test and should be operated at the specified angle towards the anchor axis. The load was applied under realistic installation conditions. In the test method the angle between the concrete specimen and the anchors was a constant 45°. Test method provides results for the diagonal force as well as for the vertical and horizontal displacement (see Figure 5).

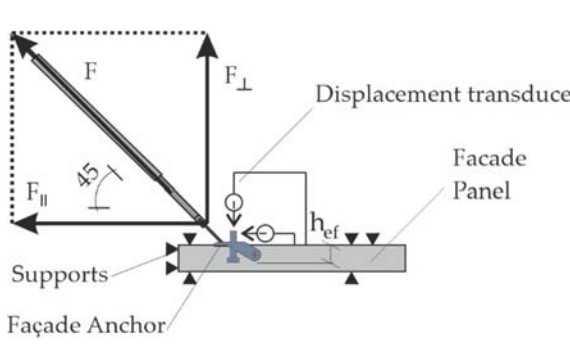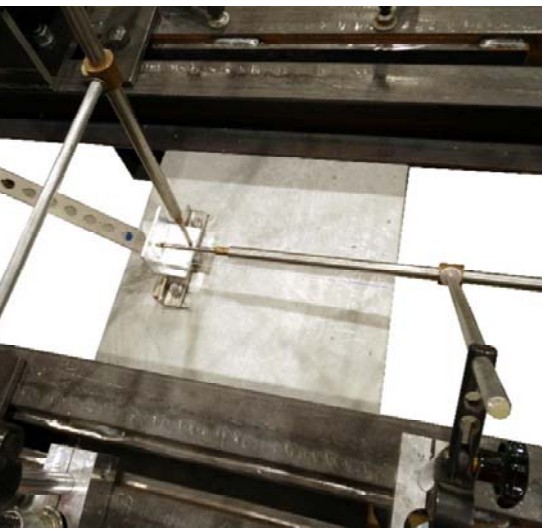

**Figure 5.** Test setup with specified angle of the anchor. The fixing place of displacement transducers and support conditions.

Figure 6 shows the specimen geometry for the testing method. With regard to façade anchors, the minimum distance to the edge of the panel is 60 mm. The displacement of the anchor relative to the surface of the test setup was >1.5$h_{ef}$ = 39 mm at a distance of 150 mm. This is the most unfavourable position of the anchor in the façade panel. The distance was measured from the axis of the anchor to the supports. The specimens were cured and stored at a temperature of 21 ± 3 °C and a relative air humidity of 60%. More details regarding the casting procedure and moulds for all specimens are presented in [19].

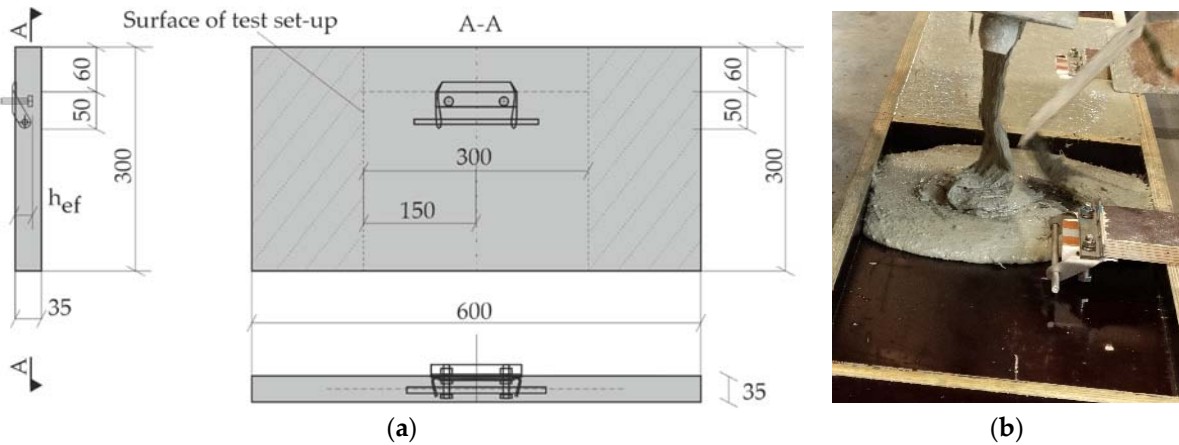

**Figure 6.** (**a**) The specimen geometry for test method. Dimensions in (mm). (**b**) Concreting process.

Figure 7 shows the experimental test setup. Displacement transducers were attached to the steel part of the support frame. They measured vertical (detail 6 in Figure 7) and horizontal (detail 5 in Figure 7) displacements. Given this test setup, the weight of the constructions had no impact on the result. The angle between the concrete specimen and the anchors was not changed. A manual hydraulic cylinder moves their piston and generates a quasi-constant load, which was applied to the end of the anchor. The load was increased in the following manner such that the maximum load was reached at 1 to 3 min after the start of the test. Unfortunately, the method of loading in pull-out tests drenders it impossible to analyse the effect of the loading speed on the load capacity of the anchors.

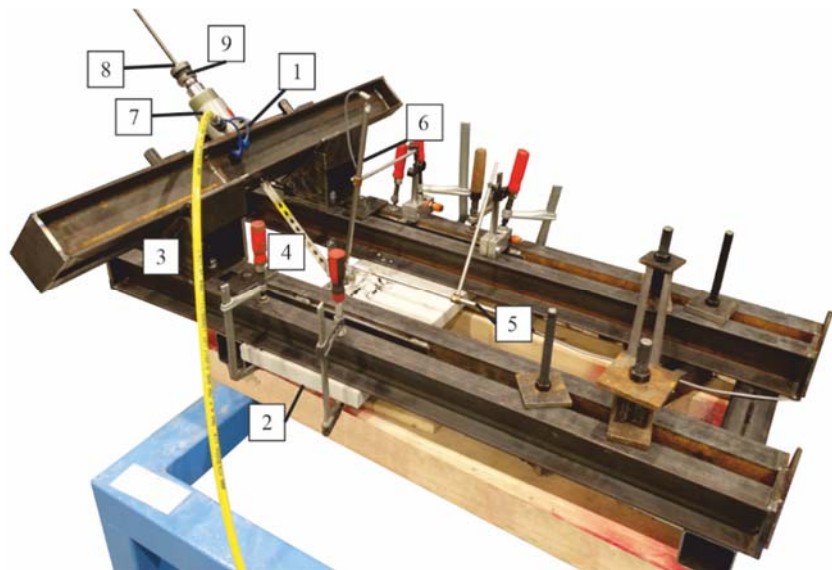

**Figure 7.** Experimental setup: 1 = point of application of symmetrical load; 2 = FRC specimen; 3 = support frame; 4 = steel anchor; 5 = horizontal displacement transducer mounted to the steel support frame; 6 = vertical displacement transducer mounted to the steel support frame; 7 = manual hydraulic cylinder; 8 = fixing point; 9 = spherical coupling.

### 3.3.2. Determination of Punching Resistance

The punching resistance of spacer bolt was determined on the basis of the experimental setup of a tension test presented in [45]. As mentioned, the differences between the ETAG and presented tests are in specimen and anchor dimensions as well in the used materials. The punching resistance was determined by means of tensile and compression tests. The specimen geometry is shown in Figure 8. The same geometry was used for tensile and

compression tests. The displacement of the anchor relative to the concrete surface was >1.5h$_{ef}$ = 37.5 mm at a distance of 150 mm. The distance was measured from the axis at the anchor head to the edge of the specimen. h$_{ef}$ is an effective anchorage depth. During the concreting process, it is desirable to distribute the fibers as well as possible all around the steel anchor. The homogeneity of the concrete mix was performed during the time of the mixing process [46]. Unfortunately, the quality control of the fibre distribution in formwork was possible only in the visual method.

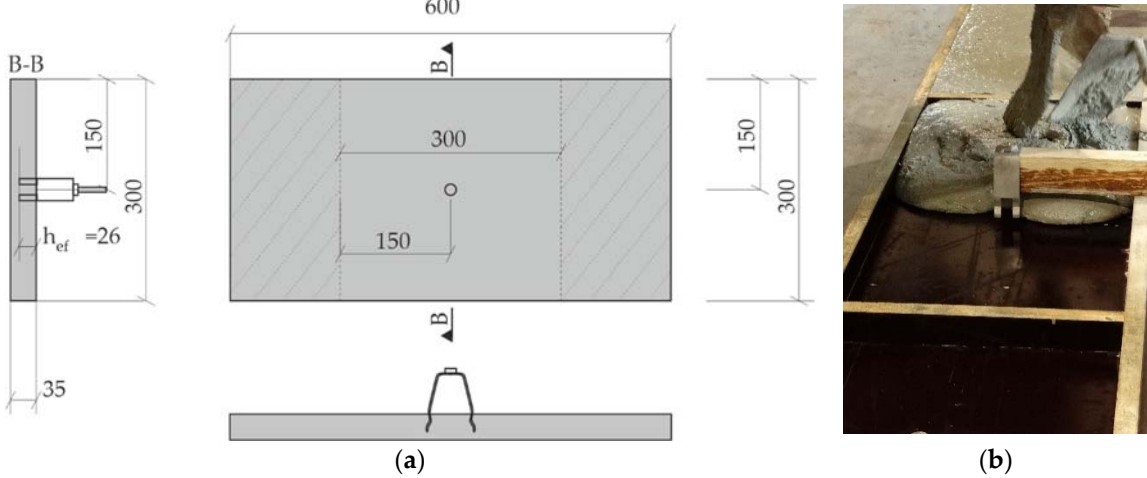

**Figure 8.** (**a**) The specimen geometry. Dimensions in (mm). (**b**) Concreting process.

The test specimens were cast horizontally. The PP fiber content of the concrete specimen was varied. The displacement during the test was measured in the vertical direction by using a displacement transducer. Once the concrete sample was cast, the anchor was connected to the test rig and exposed to load until failure. Figure 9 shows the test set-up for the pull-out test of adjustable spacer bolt. The load application was displacement controlled with the servo-hydraulic machine Schenck Hydroplus 250 kN and loading speed 3 mm/min. Two displacement transducers were mounted on both sides of the threaded rod. The distance between anchor and transducers was 100 mm > 1.5h$_{ef}$ = 37.5 mm. The distance was measured from the axis at the anchor head to the edge of the specimen.

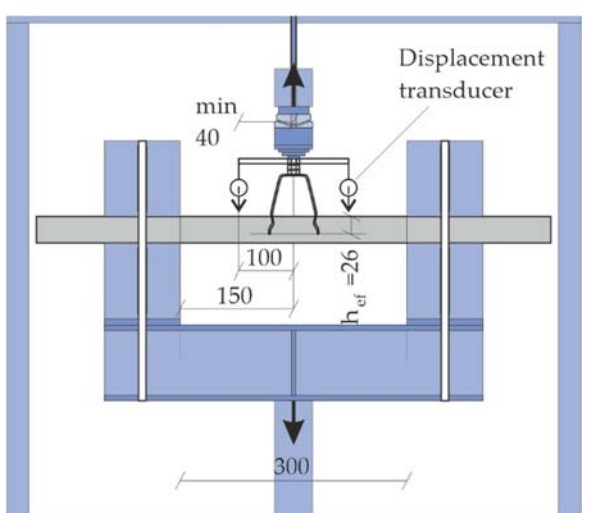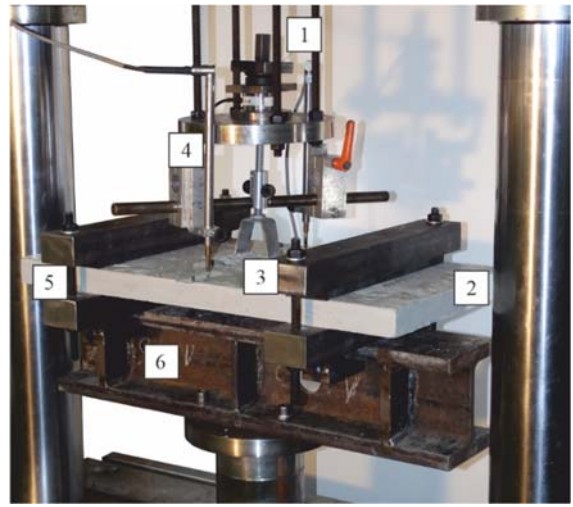

**Figure 9.** Experimental setup (pull-out test of adjustable spacer bolt): 1 = point of application of symmetrical load with the spherical coupling; 2 = FRC specimen; 3 = steel anchor; 4 = vertical displacement transducer attached to the threaded rod; 5 = tie rods; 6 = supports. Dimensions in (mm).

Figure 10 shows the test setup for the punching tests of adjustable spacer bolt. As before, the load application was displacement controlled with the loading speed 3 mm/min. A steel ring rests on the support and ensures that there are no secondary stresses due to bending in the system. Rubber pads were mounted between the concrete specimen and the steel ring. They served to ensure improved contact between the two materials. The stability of the concrete slab was guaranteed by means of the tie rods. They were fixed to the right of the supports by using a wrench. Two displacement transducers were attached directly to the threaded rod on both sides of the anchor. The deformation of the anchor is the average value from these two transducers. The distance between anchor and transducers was 100 mm > $1.5h_{ef}$ = 37.5 mm. The distance was measured from the axis at the anchor head.

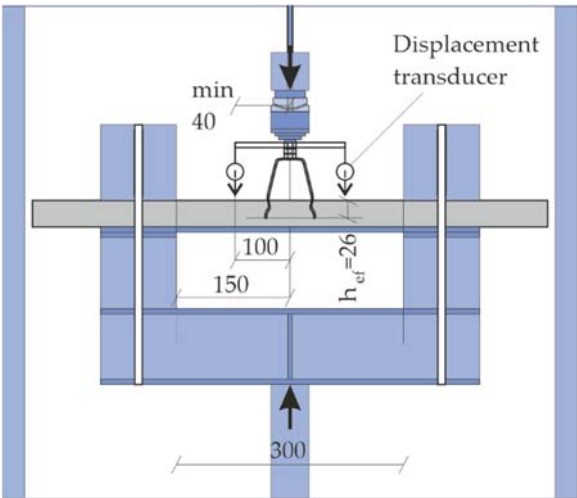 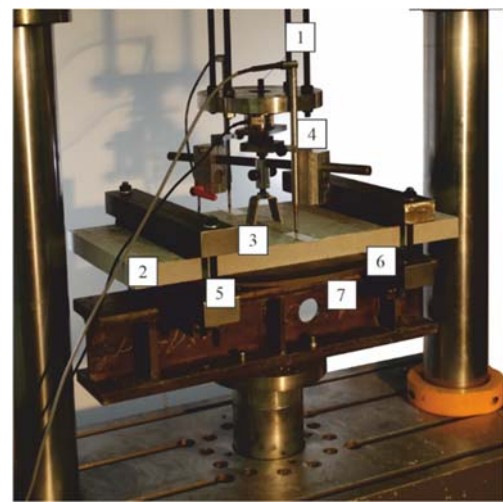

**Figure 10.** Experimental setup (punching tests of adjustable spacer bolt): 1 = point of application of symmetrical load with the spherical coupling; 2 = FRC specimen; 3 = steel anchor; 4 = vertical displacement transducer attached to the threaded rod; 5 = tie rods; 6 = supports-steel ring to ensure that there are no secondary stresses due to bending in the system; 7 = rubber pads. Dimensions in (mm).

## 4. Results

### 4.1. Results of Pull-Out Resistance Tests on Adjustable Suspended Tension-Anchor

Figures 11 and 12 presents the force-displacement curves (displacement in vertical and horizontal direction, respectively) of the tested specimens. During the test, the horizontal and vertical deformation are measured by two displacement transducers. The vertical deformation was taken to determine the point of the initial crack. In the beginning, the absorbable force increases constantly without deformation. It is well known from the previous study that the initial crack starts at the anchor head. However, that is consistent with the results in the case of the headed anchor [21]. In this article for the selected anchoring system, the moment of the first deviation from the linearity in the linear elastic response was analysed. It has been observed at a displacement of less than 0.05 mm. The value of 0.05 mm is recommended for crack mouth opening displacement in fiber reinforcement concrete [47]. In the further load phase, initial cracks forms around the anchor in the test specimens, which are visible on the concrete surface. Despite the deformation and development of the concrete cone, the anchor holds even greater forces and the resistance increases. The maximal force value differs in concrete with variable fiber dosage. At reaching this force, the concrete fails slowly due to a crack in the area close to the anchors. The test is terminated when the force goes down. The size of the concrete cone also depends on the fiber dosage in the concrete mix (see Figure 13).

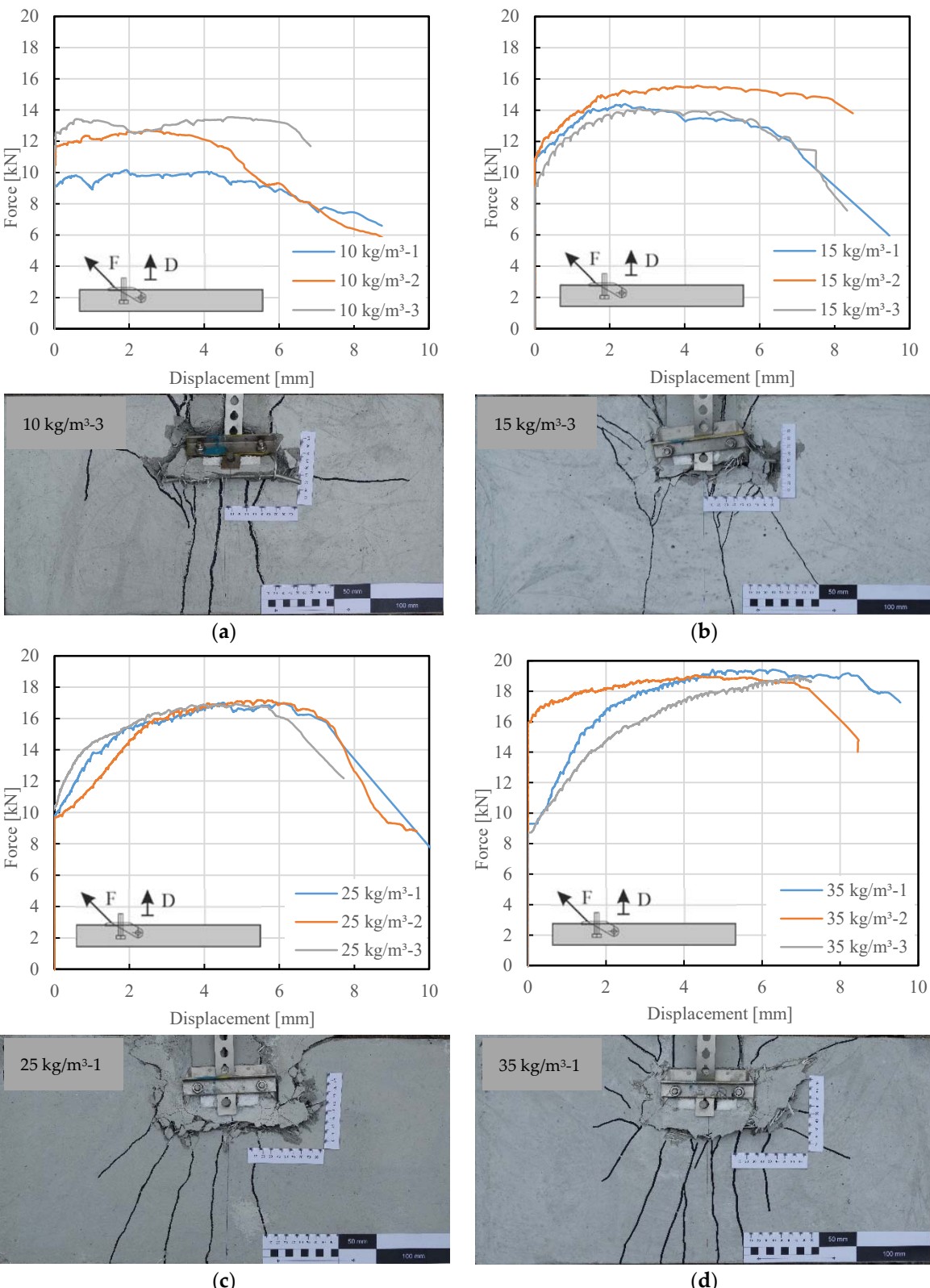

**Figure 11.** Pull-out resistance for tension-anchor in concrete with different fiber dosage. Displacement is specified in vertical direction. Top view of a sample after test with fiber dosage: (**a**) 10 kg/m$^3$, (**b**) 15 kg/m$^3$, (**c**) 25 kg/m$^3$, (**d**) 35 kg/m$^3$.

**Figure 12.** Pull-out resistance for tension-anchor in concrete with different fiber dosage. Displacement is specified in horizontal direction. Side view of a sample after test with fiber dosage: (**a**) 10 kg/m$^3$, (**b**) 15 kg/m$^3$, (**c**) 25 kg/m$^3$, (**d**) 35 kg/m$^3$.

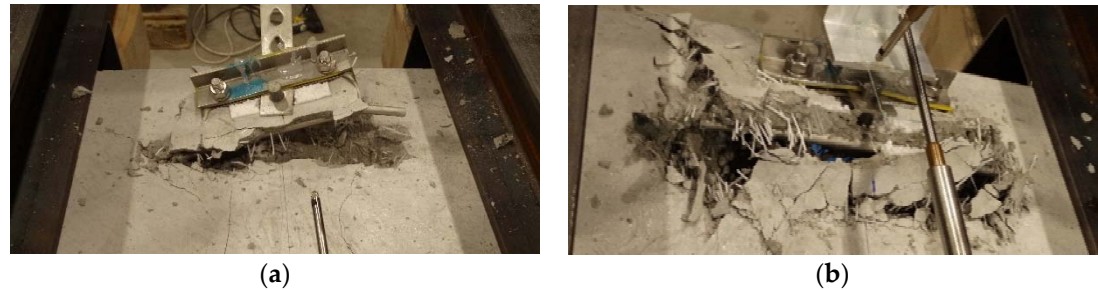

(**a**)                                                                              (**b**)

**Figure 13.** Typical failure modes of anchors: (**a**) Fiber dosage 10 kg/m$^3$. (**b**) Fiber dosage 35 kg/m$^3$.

The minimal number of the test specimens in EAD is equal to 5. Due to the limited volume of the mixer, the number of samples in tests is equal to 3. The statistical analyses were performed with Eurocode [48]. The 5% fractile of the first crack loads measured in a test series was calculated according to the statistical procedures for a confidence level of 90% according to [48]. The characteristic values were estimated with $k_s$ = 3.37. Table 5 present a comparison of maximum force and force after the first crack for different fiber dosages on characteristic and mean values.

**Table 5.** Comparison of results in pull-out tests on adjustable suspended tension-anchor.

| Mix ID | Fiber Dosages | Max. Force (st. dev) (kN) | | First Crack (st. dev) [kN] | |
|---|---|---|---|---|---|
| | | Mean Value | 5% Quantil | Mean Value | 5% Quantil |
| 1 | 10 kg/m$^3$ | 12.14 | 7.20 (0.15) | 10.44 | 6.71 (0.13) |
| 2 | 15 kg/m$^3$ | 14.69 | 12.31 (0.05) | 10.39 | 7.21 (0.11) |
| 3 | 25 kg/m$^3$ | 17.02 | 16.57 (0.01) | 9.96 | 8.86 (0.03) |
| 4 | 35 kg/m$^3$ | 19.29 | 18.60 (0.01) | 9.11 | 8.03 (0.04) |

### 4.2. Results of Pull-Out Tests on Adjustable Spacer Bolt

The force-displacement curves of the tested specimens are shown in Figures 14 and 15. The force at the anchor characterised pull-out resistance for adjustable spacer bolts. The deformation at the anchor results from the mean value of the two displacement transducers, which measures the deformation at the level of load application. During the test, the cylinder force can be continuously increased up to the first crack, which appears around the anchor. After that, a further increase in force is possible up to maximal force and corresponding deformation. Subsequently, the absorbable force drops slowly to zero.

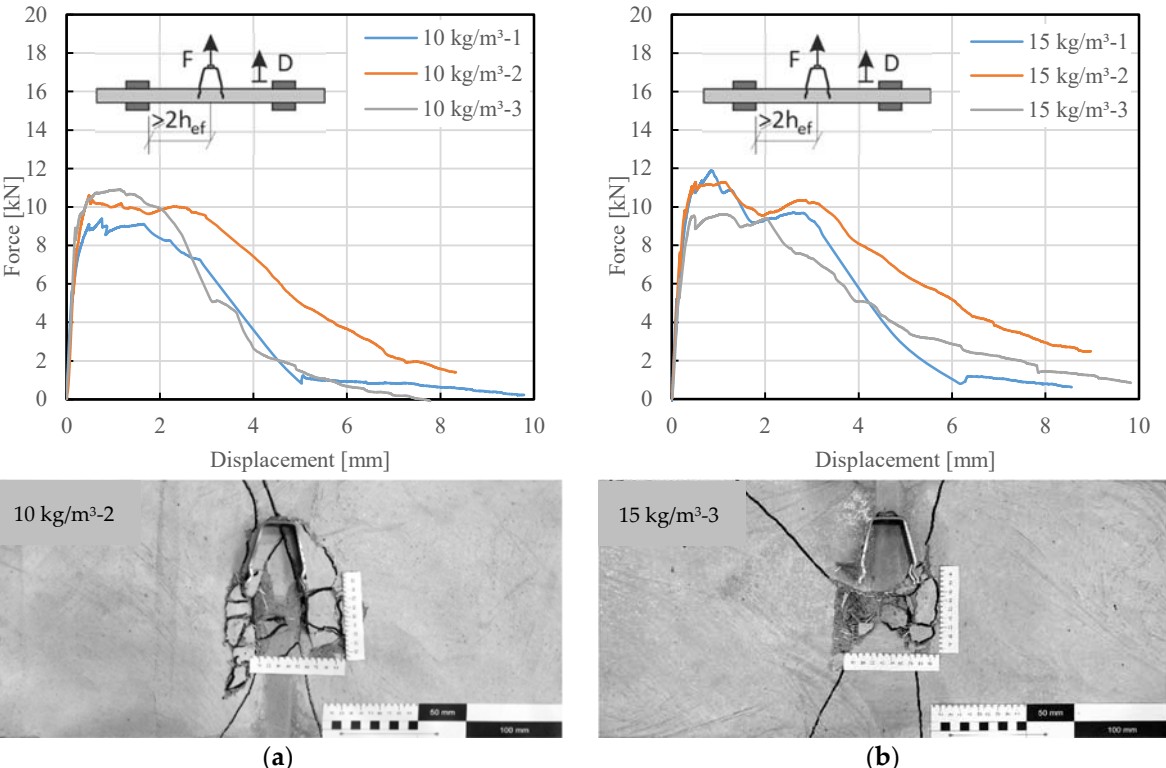

**Figure 14.** Pull-out resistance for adjustable spacer bolt in concrete with different fiber dosage. Top view of a sample after test with fiber dosage: (**a**) 10 kg/m$^3$, (**b**) 15 kg/m$^3$.

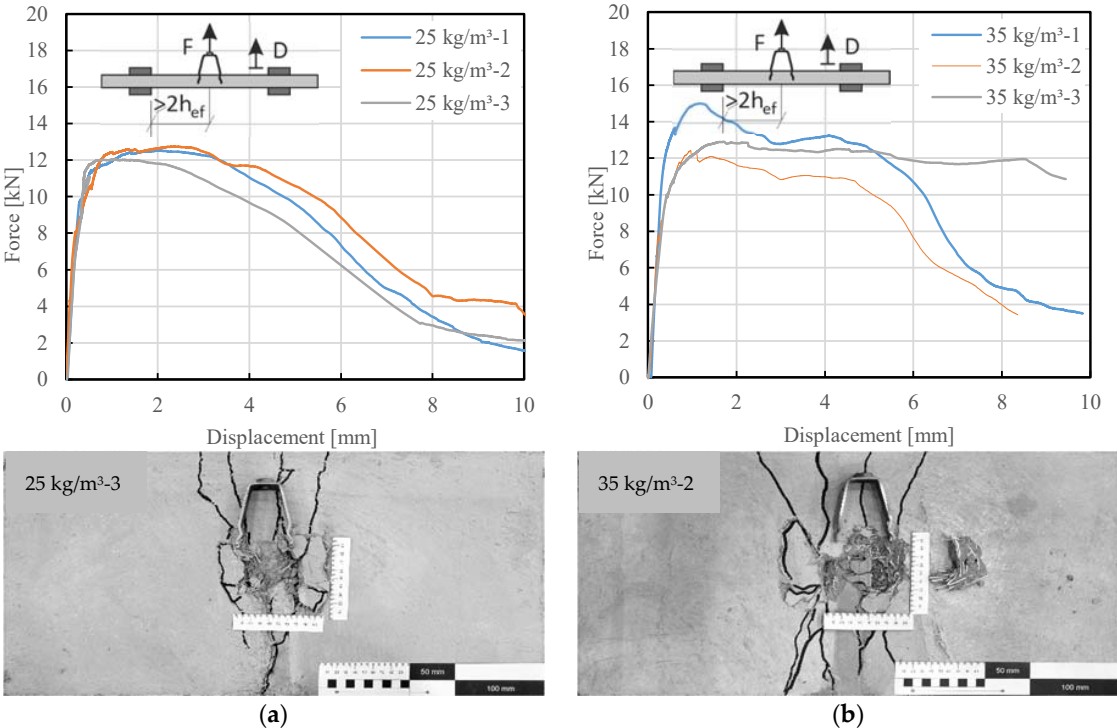

**Figure 15.** Pull-out resistance for adjustable spacer bolt in concrete with different fiber dosage. Top view of a sample after test with fiber dosage: (**a**) 25 kg/m$^3$, (**b**) 35 kg/m$^3$.

Figure 16 shows the typical failure modes of anchors in pull-out tests: concrete cone failure. The cone size is the geometrical size of the failure surface. The size of the concrete cone for the different fiber dosages did not vary much in comparison to pull-out tests on adjustable suspended tension anchor (it is within 40 mm from anchor). Table 6 presents the characteristic and mean values of maximum force and force after the first crack for different fiber dosages on characteristic values (see [48]). The test samples behave similarly despite the different fiber dosage. It depends on the characteristics of the anchor system used in tests, which are specified with the low effective embedment depth of the adjustable spacer bolt. The PP fiber accumulates in the upper part of the test specimen and there is not enough fiber reinforcement available surrounding the steel anchor. Various concrete mixtures were used to test the system behaviour; however, the first crack occurred almost simultaneously in concrete specimens (an example 8.29 kN for 15 kg/m$^3$ and 8.50 kN for 25 kg/m$^3$).

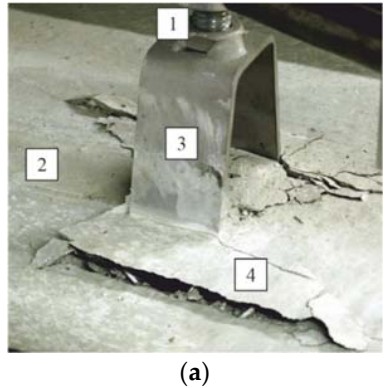

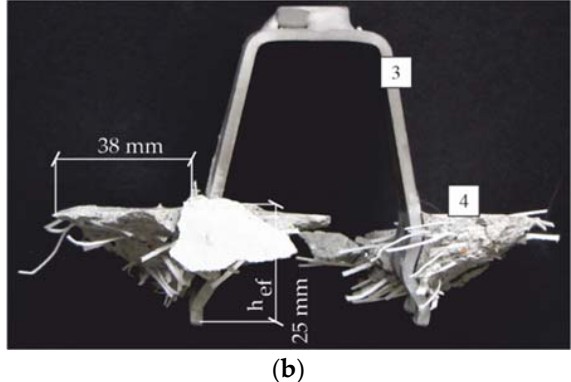

**Figure 16.** Typical failure modes of anchors in pull-out tests: (**a**) Concrete Cone failure in pull-out test after the maximum force has been reached for 15 kg/m$^3$. 1 = point of application of symmetrical load; 2 = FRC specimen; 3 = steel anchor; 4 = Concrete Cone. (**b**) Concrete Cone failure in pull-out test for spacer bolts.

**Table 6.** Comparison of results for different fiber dosages on characteristic values in pull-out tests on adjustable spacer bolt.

| Mix ID | Fiber Dosages | Max. Force (st. dev) (kN) | | First Crack (st. dev) (kN) | |
|---|---|---|---|---|---|
| | | Mean Value | 5% Quantil | Mean Value | 5% Quantil |
| 1 | 10 kg/m$^3$ | 10.29 | 7.86 (0.08) | 8.90 | 7.80 (0.04) |
| 2 | 15 kg/m$^3$ | 10.93 | 7.47 (0.11) | 9.30 | 8.29 (0.03) |
| 3 | 25 kg/m$^3$ | 12.44 | 11.25 (0.03) | 9.50 | 8.50 (0.03) |
| 4 | 35 kg/m$^3$ | 13.46 | 9.49 (0.10) | 10.20 | 9.72 (0.01) |

*4.3. Results of Punching Resistance Tests on Adjustable Spacer Bolt*

Results of punching resistance tests on adjustable spacer bolts in concrete with different fiber dosage are presented in Figures 17 and 18. For each test specimen, the absorbable force increases constantly. Cracks appear slowly on the lower concrete surface. The crack width increases and results in breakage or debonding of the fiber in concrete structures. Some of the fibers reached their tensile strength during the load tests. The cylinder force increases up to the maximum load, which depends on fiber dosage. At the maximal force, the test specimen fails in the area of the anchors. The fiber dosage influences the maximum force in punching tests. For the maximum fiber content of 35 kg/m$^3$, not only was the maximum force affected but the post-crack behaviour was also influenced. The higher ultimate load capacity of the concrete cone results in a higher safety factor for the concrete façade elements at constant design loads.

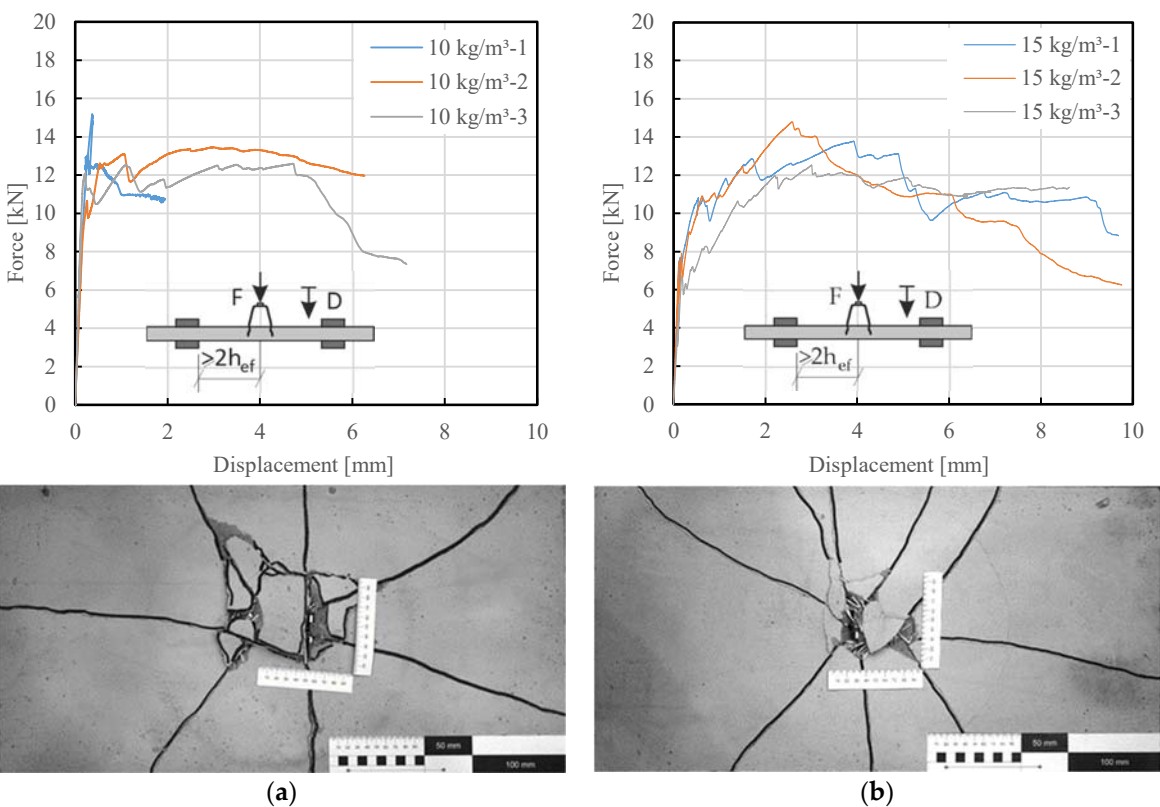

**Figure 17.** Results of punching resistance tests on adjustable spacer bolt in concrete with different fiber dosage. Top view of a sample after test with fiber dosage: (**a**) 10 kg/m$^3$, (**b**) 15 kg/m$^3$.

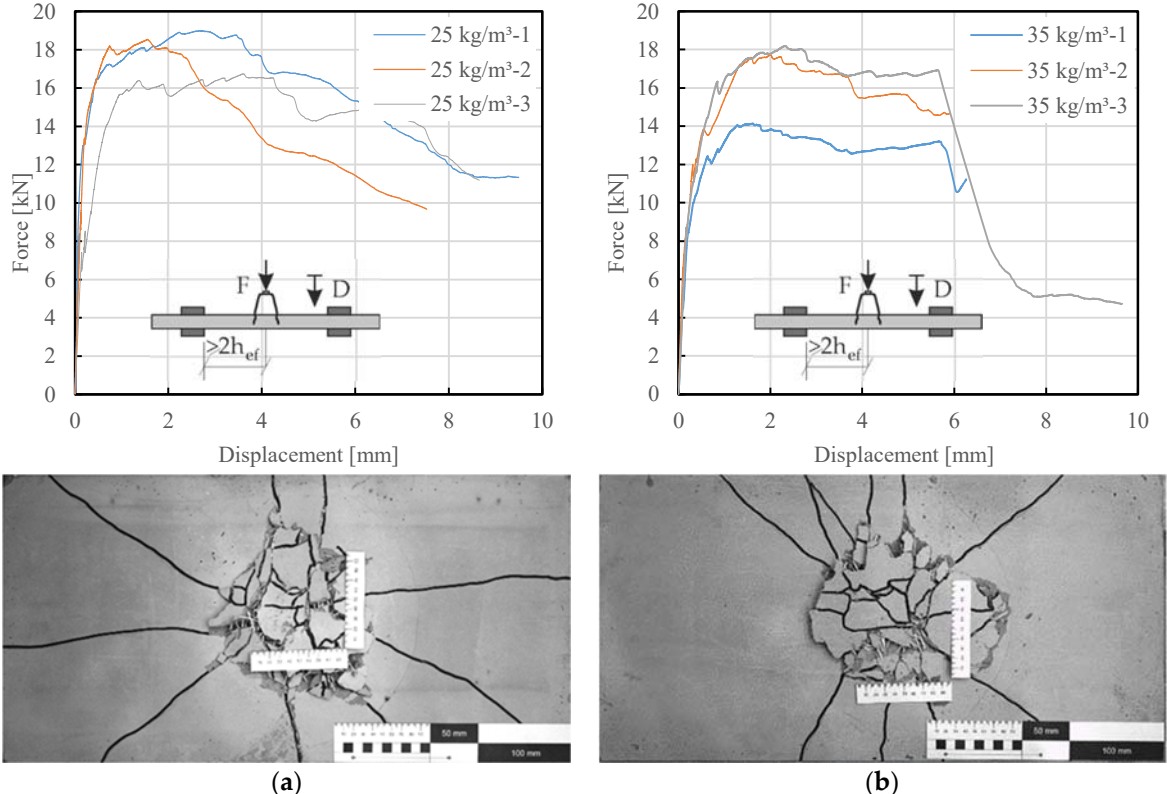

**Figure 18.** Results of punching resistance tests on adjustable spacer bolt in concrete with different fiber dosage. Top view of a sample after test with fiber dosage: (**a**) 25 kg/m$^3$, (**b**) 35 kg/m$^3$.

The characteristic and mean values were estimated according to [48] and presented in Table 7. The analysis of results showed that the different fiber dosage influences the effective fiber orientation. For panels with lower fiber content, the possibility of ineffective fiber distribution around the anchors was increased.

**Table 7.** Comparison of results for different fiber dosages on characteristic values in punching tests on adjustable spacer bolt.

| Mix ID | Fiber Dosages | Max. Force (st. dev) (kN) | | First Crack (st. dev) (kN) | |
|---|---|---|---|---|---|
| | | Mean Value | 5% Quantil | Mean Value | 5% Quantil |
| 1 | 10 kg/m$^3$ | 13.74 | 9.40 (0.09) | 9.00 | 3.30 (0.30) |
| 2 | 15 kg/m$^3$ | 13.70 | 10.30 (0.08) | 6.00 | 4.96 (0.07) |
| 3 | 25 kg/m$^3$ | 18.09 | 14.40 (0.07) | 9.00 | 7.09 (0.07) |
| 4 | 35 kg/m$^3$ | 17.96 | 10.40 (0.14) | 11.20 | 9.44 (0.05) |

Figure 19 shows the crack development in the concrete specimen during a punching test. The first crack formed at the lower part of the concrete specimen near the steel anchor. The tensile stresses in the concrete are distributed for further distances and influences a larger radius of concrete cone (compare Figure 19a,b).

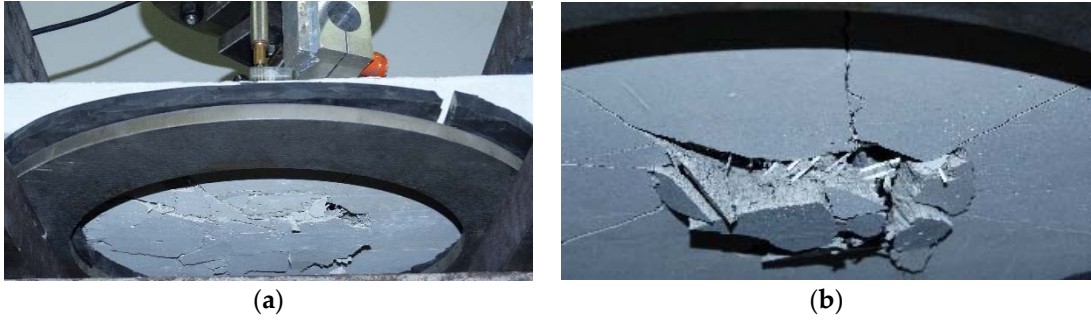

<center>(<b>a</b>) (<b>b</b>)</center>

**Figure 19.** The failure of the concrete cone with splitting cracks for (**a**) 35 kg/m$^3$ and (**b**) 15 kg/m$^3$.

## 5. Discussion

Determination of pull-out and punching resistance is clarified in accordance with [49]. The adjustable spacer bolts by their double sleeve construction transfers the horizontal loads as an anchor group (factor $\psi_{ec,N}$ in Equation (2)). In comparison to the study of Cattaneo and Muciaccia [50], fiber dosage does not result in steel failure in tests. All experiments occur with cone failure. In the case of concrete cone failure, the partial safety factors $\gamma_{Mc} = \gamma_c \cdot \gamma_{inst}$ are calculated using Equation (1).

$$N_{Ed}^g \leq N_{Rd,c} = \frac{N_{Rk,c}}{\gamma_{Mc}} \tag{1}$$

The characteristic force $N_{Rk,c}$ depends on concrete properties and anchor geometry (see Equation (2) with factor $k_{ucr,N}$ = 12.7 for uncraced concrete). For variable fiber dosage, the compressive strength of concrete $f_{ck,cube}$ and the size of the concrete cone with the surface area $A_{cr,N}$ are different. The surface area was furthermore adapted to the test method. For pull-out resistance, tests on adjustable suspended tension-anchor and punching resistance differences were higher than for the pull pull-out tests on the adjustable spacer bolt. The calculation depends on effective anchorage depth $h_{ef}$ (factor $\psi_{re,N}$) and from the location of the anchor (factor $\psi_{s,N}$ and $\psi_{M,N}$). For the adjustable suspended tension-anchor, the distance from the edge of the sample $c_1$ = 60 mm was also considered. The following equations was used.

$$N_{Rk,c} = k_{ucr,N} \cdot \sqrt{f_{ck,cube}} \cdot h_{ef}^{1.5} \cdot \frac{A_{cr,N}}{A_{cr,N}^{one}} \cdot \psi_{s,N} \cdot \psi_{re,N} \cdot \psi_{ec,N} \cdot \psi_{M,N} \tag{2}$$

### 5.1. Pull-Out Resistance of Adjustable Suspended Tension-Anchor

Figure 20 shows the pull-out resistance of anchor in concrete with varying fiber content. Important information that is relevant for HPFRC and for the load capacity of the anchors is the development of the first crack in the specimen, which appears in the initial phase of loading. The maximum load-bearing capacity and dimensions of façade elements can be calculated on the basis of the maximum forces. The calculation results are comparable to the first crack in pull-out test because the maximal force in tests was much higher.

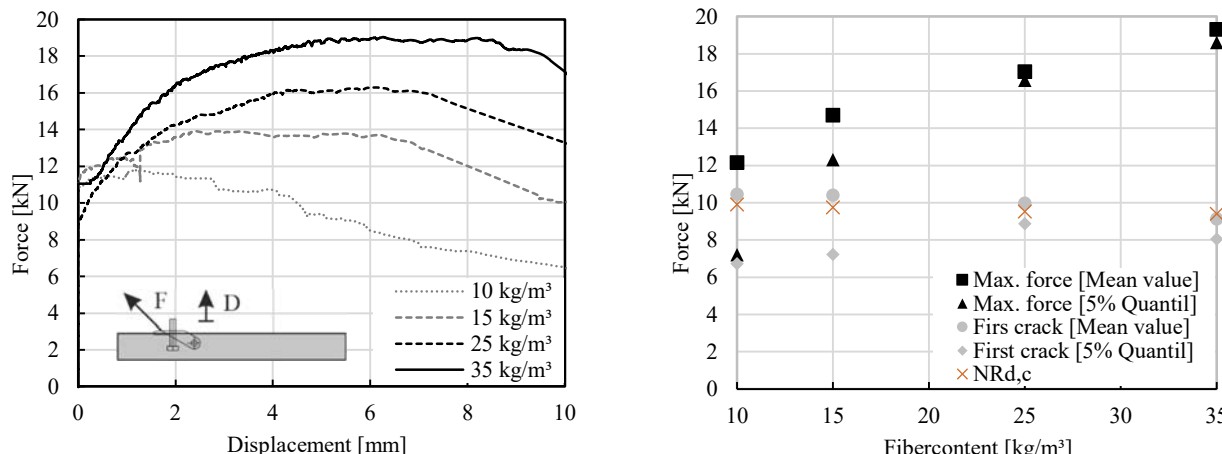

**Figure 20.** Pull-out resistance of anchor in concrete with different fiber dosages of MasterFiber®235 (PP).

Figure 21 shows the crack development in the concrete specimen. The first crack forms near the steel anchor. The angle of crack is equal to 35°. At the end of the test, the failure of the concrete cone was clearly visible. Similar results were observed on the experimental tests with steel fibers by Kurz et al. [21] when investigating the load bearing capacity of steel dowels in the HPSFRC.

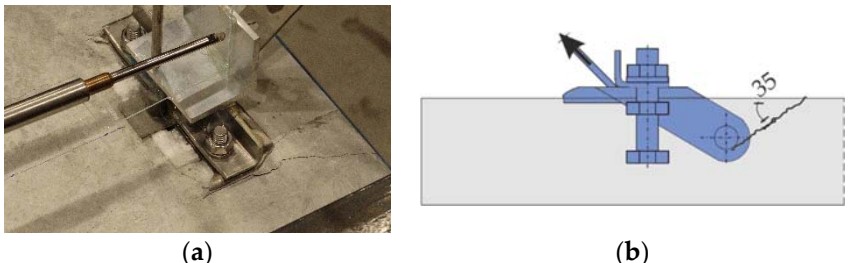

(**a**) (**b**)

**Figure 21.** (**a**) Crack development in the experimental test. (**b**) Scheme of crack development.

However, micro-crack formation and subsequent exposure to moisture and temperature can often result in crack formation in these components. In panels reinforced with steel rebars, this can cause corrosion formation, resulting in unwanted concrete spalling and cosmetic colour changes of the façade panels. A possible solution to this problem is the higher steel reinforcement grade or to dispense with passive reinforcement and replacing with fibers. Reducing conventional steel reinforcement simultaneously minimises the concrete cover in the elements. By eliminating the concrete cover, the façade panels can be made more filigree. Thanks to the synthetic fibers used, corrosion formation or colour changes of the façade panels are not possible. Furthermore, the polymer fibers embedded in the concrete matrix have a crack-inhibiting effect, which permanently guarantees the optical properties and durability of concrete façades [51].

Figure 22 shows the typical failure modes of anchors with higher fiber dosage. On the left of the picture, a completely inefficient fiber orientation is depicted (Figure 22a). This fiber orientation results in a faster failure of the façade panel when the first crack occurs in the concrete. The inefficient fibers do not guarantee the safety of the concrete elements. On the right of Figure 22, an ideal fiber orientation is depicted (Figure 22b). Given this orientation of the fibers, the first crack in the concrete exploits the fiber stress properties. The ideal fiber orientation increases the resistance of the steel anchor. A comparison with high fiber dosage on the Figure 22 and low fiber dosage can be seen in Figure 23. This is also seen in two different specimens with a small fiber content of 10 kg/m$^3$ and a larger fiber content of 25 kg/m$^3$ (see details on HPFRC failure in Figure 24). The higher fiber content causes the fibers to be more evenly distributed around the steel anchors. The correct

arrangement of the fibers around the steel anchor improves the load-bearing capacity of the HPFRC elements, which can be observed through the formation of cracks between the anchor and free edge of the concrete sample. Nilforoush et al. [52] observed the same additional radial bending cracks, which form between the free edge of concrete slabs and anchor bolt (see A-A in Figure 22). It is well known from the tests on mechanical properties that the high dosage also causes strength reduction.

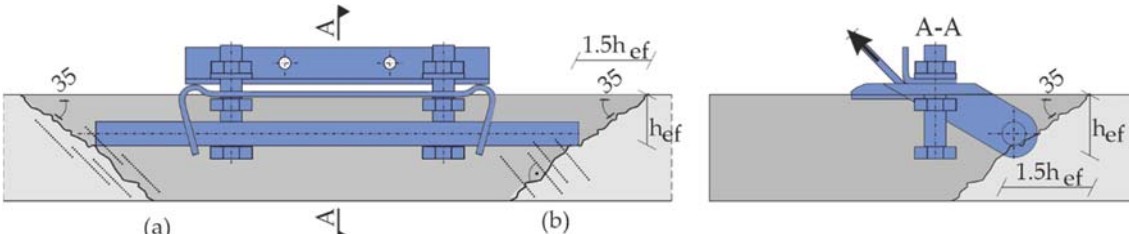

**Figure 22.** Typical failure modes of anchors with higher fiber dosage: (**a**) Inefficient orientation of fibers. (**b**) Ideal orientation of fibers.

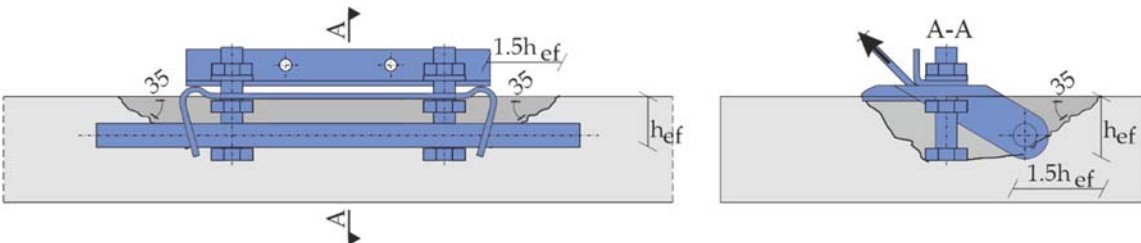

**Figure 23.** Typical failure modes of anchors in concrete with lower fiber dosage.

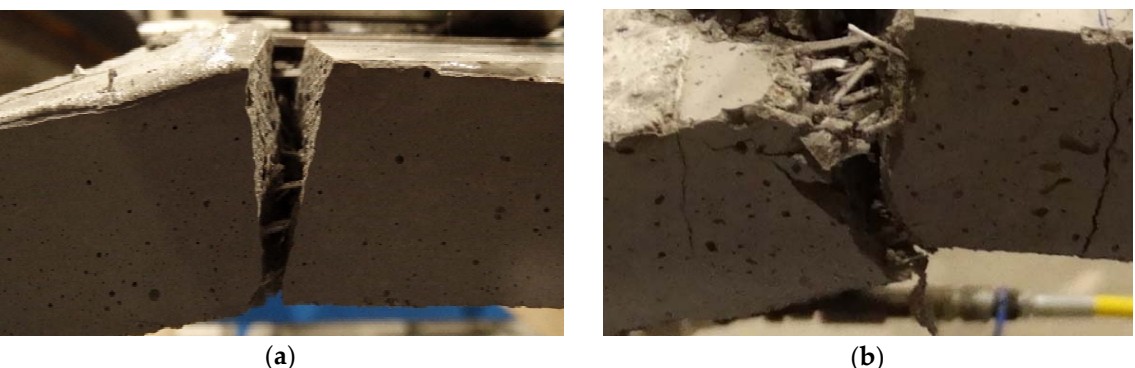

**Figure 24.** Details of HPFRC failure: (**a**) Example for inefficient fiber orientation (10 kg/m$^3$). (**b**) Example for ideal fiber orientation (25 kg/m$^3$).

### 5.2. Pull-Out Resistance of Adjustable Spacer Bolt

The maximum tensile force defines the maximum wind suction load per horizontal anchor. The experimental results are shown in Figure 25. The results are presented as characteristic and mean value. The highest PP fiber content examined in the concrete composition amounted to 35 kg/m$^3$. However, the dosage of more than 25 kg/m$^3$ of fibers to the concrete mix has less impact on the tensile strength of anchors in the concrete façade element. The comparison of the force-displacement curves for the dosage of 25 kg/m$^3$ and 35 kg/m$^3$ of fibers reveals that the cracking behaviour of concrete for these two fiber contents do not differ significantly. The clear differences are in the first crack of concrete samples. Comparable results were obtained with a fiber dosage of 15 kg/m$^3$ and 10 kg/m$^3$.

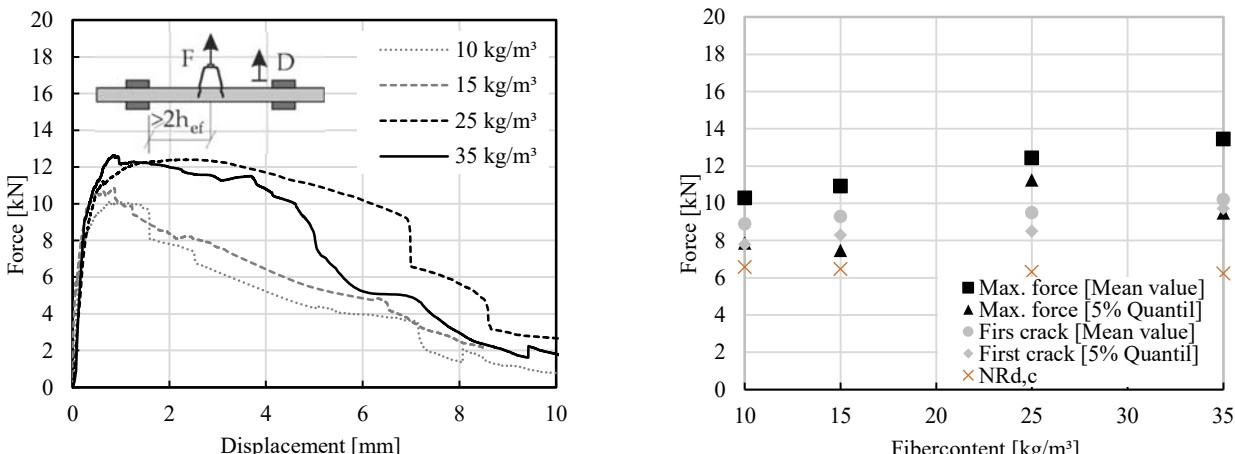

**Figure 25.** Results of tensile strength test for concrete with different fiber dosages of MasterFiber®235 (PP).

Figure 26 shows the crack development in the concrete specimen during a tensile test. This principal sketch shows the failure of anchors during tensile tests. The first crack forms near the steel anchor. The angle of the crack is 35°. When the experiment was completed, the shape of the concrete part around the anchor resembled the failure of the concrete cone. The diameter of the concrete cone failure was irregular. In the other point the distance was about $1.5h_{ef} = 1.5 \times 25$ mm $= 37.5$ mm. Differences in the size of the concrete cone between specimens with differing fiber content were not observed.

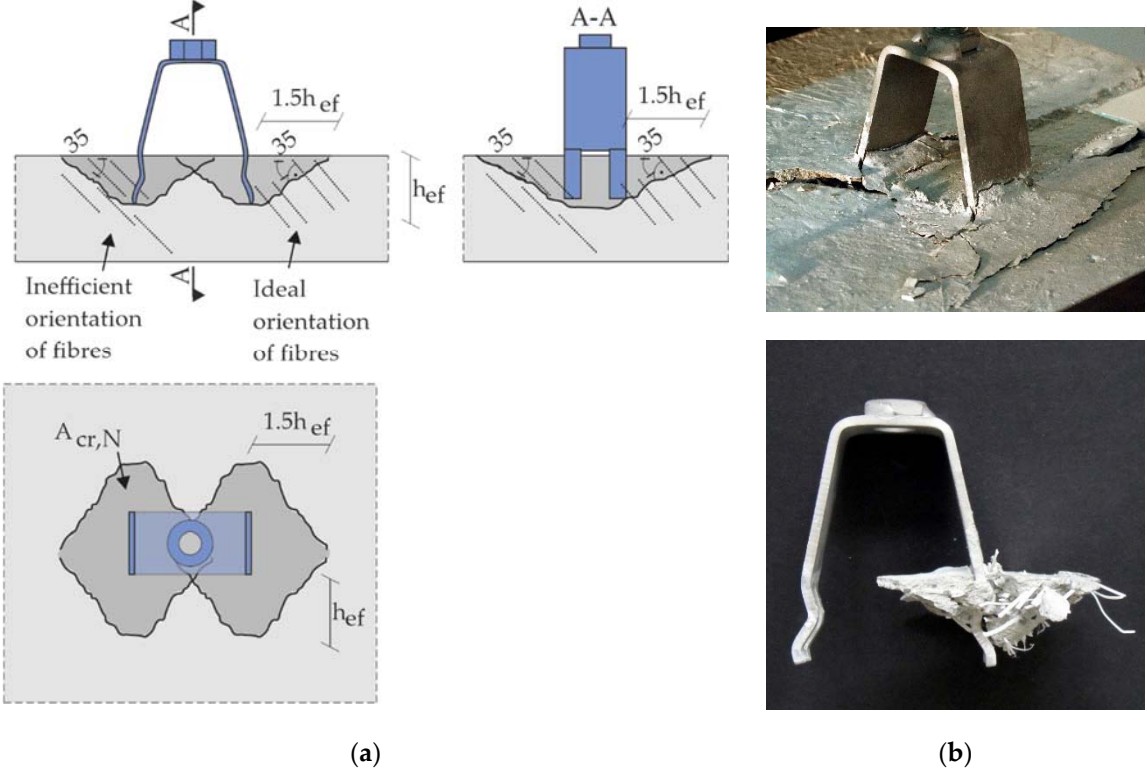

(**a**) (**b**)

**Figure 26.** Typical failure modes of anchors in pull-out tests: (**a**) Principal sketch of inefficient and ideal orientation of fibers. (**b**) Concrete Cone failure in pull-out test for spacer bolts.

### 5.3. Punching Resistance of Adjustable Spacer Bolt

The maximum compressive force defines the maximum wind pressure per horizontal anchor. Figure 27 shows the results of the punching resistance tests. The highest PP fiber

content examined in the concrete composition amounted to 35 kg/m³. The dosage of more than 10 kg/m³ of fibers to the concrete mix has an understandable impact on the first crack force in punching tests. This also affects the punching resistance of the concrete element. The maximum force in tests has less impact in comparison with the force that results from the first crack of the element. A comparison of the force-displacement curves for the dosage of 25 kg/m³ and 35 kg/m³ of fibers reveals that the cracking behaviour of concrete for these two fiber contents does not differ significantly.

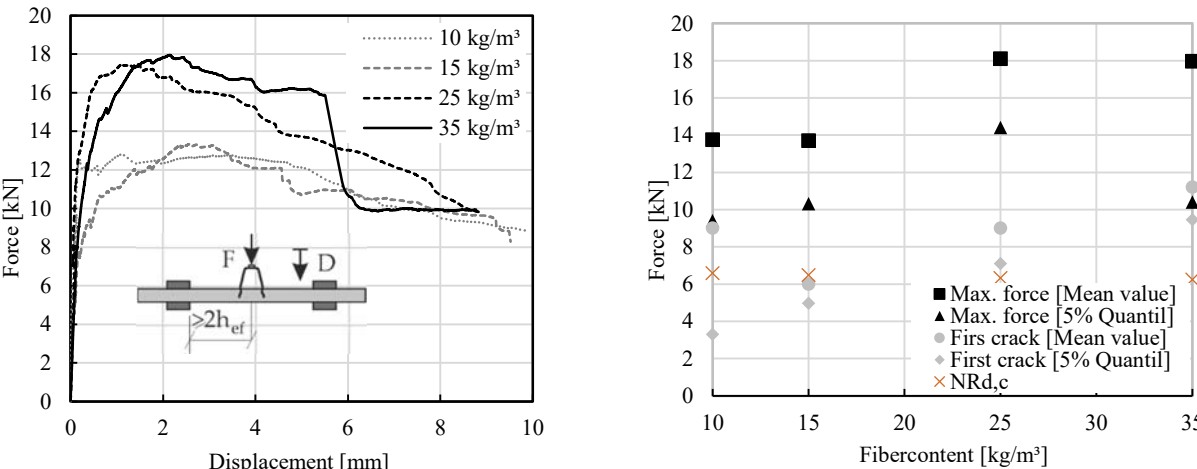

**Figure 27.** Results of the punching resistance test for concrete with different fiber dosages of MasterFiber®235 (PP).

Figure 28 shows the crack development in the concrete specimen during a punching test. This principal sketch shows the failure of anchors during the investigations. The angle of the crack is 35°. When the experiment was completed, the shape of the concrete part around the anchor resembled the failure of the concrete cone. The diameter of the concrete cone failure was irregular by the occurrence of numerous splitting cracks. In the further point the distance was about 1.5$h_{ef}$ = 1.5 × 25 mm = 37.5 mm. The size of the cone depends on the fiber dosage in the concrete mix. With 15 kg/m³, the concrete cone was only limited to 100 mm. With a fiber content of 35 kg/m³, the size of the cone exceeds 150 mm. This is due to the inclusion of the fibers in the punching resistance test after the first crack. The higher fiber dosage activates more concrete volume and, at the same time, more fiber.

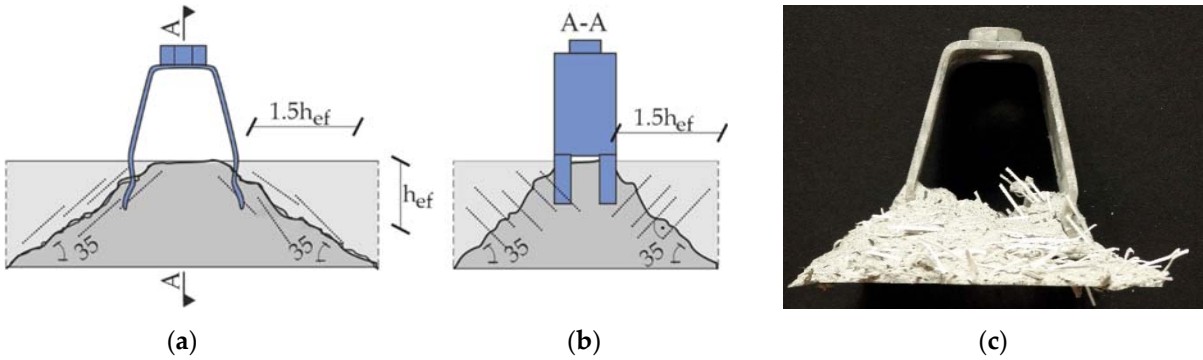

| (a) | (b) | (c) |

**Figure 28.** Typical failure modes of anchors in punching tests: (**a**) Inefficient orientation of fibers. (**b**) Ideal orientation of fibers. (**c**) Concrete Cone failure in punching test.

## 6. Conclusions

Two different anchors were investigated within the scope of this study. The adjustable suspended tension anchors were examined in pull-out tests. The adjustable spacer bolts

were examined in pull-out and punching tests. This study allows for the following conclusions to be drawn in the area of system behaviour:

- The pull-out resistance of adjustable suspended tension-anchor is affected by different fiber dosages. The higher the fiber content, the greater the force was at the moment of the first deviation from the linearity in the linear elastic response. The force increases from 6.71 kN for 10 kg/m$^3$ to 8.86 kN for 25 kg/m$^3$. In general, it can be stated that fibers have a positive effect in the concrete's post crack behaviour. It has been shown for the samples with fiber dosage 10 kg/m$^3$ that the maximal force reaches the value of 7.20 kN and, for the samples 35 kg/m$^3$, the value of 18.60 kN is reached.

- Comparing the results of pull-out tests on adjustable spacer bolt shows an influence of the varying fiber dosage. With a fiber addition of 10 kg/m$^3$ and 35 kg/m$^3$, the maximum force was varied between 10.29 kN and 13.46 kN. The first higher increment of deformation in time increment varied between force 7.80 kN and 9.72 kN for fiber additions between 10 kg/m$^3$ and 35 kg/m$^3$. Therefore, experimental tests show that the amount of fibers does not significantly influence the post-crack behaviour of concrete in pull-out resistance of adjustable spacer bolt.

- The punching resistance of adjustable spacer bolt is strongly influenced by the fiber dosage in the concrete mix. The first deviation from the linearity in the linear elastic anchor behaviour for the fiber addition of 35 kg/m$^3$ is equal to force 9.44 kN and is greater than force 3.30 kN for the specimen with a fiber addition of 10 kg/m$^3$. The correlation of the force-displacement curves shows that with a fiber addition of 25 kg/m$^3$ and 35 kg/m$^3$, the results are similar. The maximal force is 18.09 kN and 17.96 kN, respectively. Differences in punching resistance can also be seen in the magnitude of cone failure. With a fiber content of 35 kg/m$^3$, the fractured part of the concrete is formed in a larger specimen area than with a fiber content of 15 kg/m$^3$. Due to the higher fiber dosage, larger volumes of concrete are integrated and the diameter of concrete cone around the anchor increases.

The potential application of research results in engineering fields is with respect to precast concrete systems. The façade cladding as a product group has an optimisation potential in the concrete industry. It is of interest to improve the capacity and economic efficiency of these non-structural elements. That will happen through research and analysis of HPFRC and the behaviour of anchors in this material.

**Author Contributions:** Conceptualization, S.G., M.P. and M.S.-C.; investigation, S.G., M.P. and M.S.-C.; writing—original draft preparation, S.G., M.P., M.S.-C. and N.S.B.; supervision, S.G., M.P., M.S.-C. and N.S.B. All authors have read and agreed to the published version of the manuscript.

**Funding:** This research was funded by Master Builders Solutions Deutschland GmbH.

**Institutional Review Board Statement:** Not applicable.

**Informed Consent Statement:** Not applicable.

**Data Availability Statement:** Not applicable.

**Acknowledgments:** This research work was supported by Master Builders Solutions Deutschland GmbH. The authors would like to thank staff of Civil Engineering Department of Technical University in Kaiserslautern for their support extended during the experimental works carried out in the laboratory.

**Conflicts of Interest:** The authors declare no conflict of interest.

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
