# Peer review of "Influence of Different Fiber Dosages on the Behaviour of Façade Anchors in High-Performance Concrete"

_2673-4109, doi:10.3390/civileng2030031_

Round 1

Reviewer 1 Report

The reviewer would like to thank the authors for their efforts to improve the current knowledge of the influence of the finer dosage on the anchor performance. The aim of this experimental program was to analyse the impact of added fibers in the concrete composition on the behaviour of anchors using the adequate test setups.

In general, the manuscript is organized well but lacks number of very important details and have some technical and grammatical flaws which are mentioned below. 

The manuscript in its current shape can not be accepted for the publication and requires major revision. The authors are encouraged to incorporate comments and feedback in order to further improve presented study, and furthermore to be considered for publication. 

Please find my comments:

Line 51: check language please: “.. to increases of the…”

Line 55: check language please: “…save…”. Please use safe.

Line 107: Better explanation of the components is required for the reader’s better understanding, and updated Fig.2.

Line 124: What is meant by no risk of injury? The sharpness of the steel fibers? If so, then please modify the sentence.

Line 134, Table: It is unclear how are the mechanical properties of fibers identified. Whether they experimental characterized within the experimental work or provide by the producer? Please updated the missing information.

Line 145: It is written: …three different concrete mixtures…

Furthermore, Table 3. shows 4 different concrete batches? Where there 3 or 4 concrete mixtures?

Line 160: I believe that cube specimens were stored in lime-saturated water or in a storage room with >95% of relative humidity, and the storage conditions were the same for all the concretes Please add explanation on the storage conditions (RH and T information), and specify the exact age of concrete when tests were carried out.

It says only after 30 days, but it is important to know if all cubes where tests at exactly the same age or one concrete batch was tested at e.g. 31 days, and other at 32 days.

Additionally, please explain a bit more on casting conditions (e.g. outdoor, indoor) and also for how long specimens were kept in the moulds. When comparing mechanical properties (e.g. compressive strength) between different concrete mixtures procedure consistency during the casting, storing and testing is of high importance for obtaining reliable results.

Please consider above mentioned comment in the manuscript.

Figure 4: Clearly the mean values are plot with corresponding scatter band, that I assume it is COV? It would be interesting to list the values, such that minimum and maximum values are more clearly presented.

Additionally, any information about high scatter bend in case of concrete ID 3? What is the expected range of scatter in those type of tests?

Line 162-168: Better explanation is required on the concrete strength and fiber dosage effect. Please update the chapter with more literature evidence? It is not clear for a reader whether this trend is expected or not?

Is the message that for the normal strength concretes (under 30 MPa) the higher dosage will increase compressive strength, but the opposite trend is valid for higher strength concretes (higher dosage leads to lower fc)?  If so, what is the concrete class where this trend changes, or in the other words what is the limit of concrete class  that shows increased fc with increased fibers dosage?

Please revise the chapter since it requires significant improvements.

Figure 6b: Please separate it and put casting Photo at the appropriate place. I suggest to even add short chapter e.g. “Casting and specimens preparation” where you can provide all details regarding the casting procedure and moulds for all specimens.

Line 190: Please check previous comment regarding Line 160. Is the statement in Line 190 valid also for the cubes tested to identify compressive strength?

Figure 7: The test setup requires more detailed explanations, and I suggest to add 1-2more detailed photo/sketch in order that someone is able to reproduce those tests.

For example it is not easy to understand where and how exactly the displacement transducers are set, and what exactly they record (e.g. displacement. relative to the plate?)

Line 199: How was the load application controlled? Force controlled mode? How and was the same loading ramp ensured for the each test?

Any thoughts about presence of the loading rate effect considering the possibility that one anchor (test) was loaded 60s until the peak was reached, and the second anchor (test) 180s? Please add explanation on that.

Chapter 3.2.2.: How was the load applied in case of pullout and punching tests? I assume it was displacement controlled load application? If so, please report the loading speed.

Figure 12: I would like to see more zoomed plot with the x-axis limit 0-3mm displacement.

In this way it is not possible to see the part and expected deformations during the loading.

Line 142: How can be that you have increased force even up to 10kN without deformations?

Figure 12: I suggest to make mean curve for each of 4 concretes and plot only mean curves in a one Figure. In this way it is more difficult for a reader to make real comparison in anchor behavior regarding the stiffness changes. Additionally, it is important and interesting for the study to compare stiffnesses in initial part (loading part), both for all tests for each concrete, and mean values among concretes.

Table 5: Any reason why results are presented as characteristic values? Why mean values are not used with corresponding coefficient of variation (COV)?

Line 259: The first crack is not the one that is visible on the concrete surface. It is well known from the previous studies that the initial crack start at the anchor head. Therefore, I it not fully correct to write first crack, rather than used first visible crack on the concrete surface.

Additionally, it would be very interesting if you could mark the exact position in the Force-Displacement diagram when the first visible crack on the surface was observed.

If you want to estimate the moment of the first crack, than at least the linear elastic response should be checked and see the moment of the first deviation from the linearity.

Line 267: Can you better document how it is proven that the concrete cone did not vary for the different fiber dosages?

Table 6: Again, why there is a need to work with characteristic values since we have mean experimental values? Please use experimental values as they were experimentally obtained and make comparison.

Figure 15: Please see my comment as earlier regarding the plot and making the comparison.

Line 286: What is meant by: “…not only the maximum force was achieved…”?

Line 287: Can you better describe this sentences: The higher load capacity of steel anchors means a safety factor for the concrete facade elements.

Why that if the concrete cone capacity occurs first?

Table 7: As earlier, work only with mean values, no need to use characteristic values and COV for that values!

Figure 16: It is not pure cone failure, since there are a lot of splitting cracks. Interestingly, you are saying the cone failure and then reporting crack close to the anchor?

Any explanations on that? Are the splitting cracks results of the setup imperfections (e.g. presence of eccentricity) or it is result of the concrete geometry (e.g. member thickness).

Additionally, how did you deal with the eccentricity while loading in the setup for the pullout tests? Was the spherical coupling used?

Please describe that better in the manuscript.

Line 312: It is written: ‘…size of the concrete cone with the surface area Acr,N are different...”

Is not that in conflict with you previous statement that concrete cone does not vary? (Line 267).

Line 317: Now it is reported that the crack forms at the initial phase of loading? What is that point in the load displacement diagram? Please show the zoomed view of loading part.

Line 309: Please show exact equation of NRk,c and how the calculation is done.

Figure 21: Clearly more fibers lead to a higher possibility of the “correct” fiber orientation that is beneficial for a give failure mode. However, you need also to account for you earlier statement that high dosage also causes strength reduction. Please support better you assumptions and conclusions here.

Conclusions need to be critically revised especially regarding the use of the term “first crack”, and use comparison of the mean maximum values.

Reviewer 2 Report

The article covers the topic of the Influence of Different Fiber Dosages on the Behaviour  of Façade Anchors in High-Performance Concrete.
The manuscript has good cohesion. This is an interesting paper that deals with a timely topic and novel idea. 
The topic of the paper suits this journal. 

However the following modification should be considered to improve the quality of paper:

1. It is worth to add graphical abstract.
2. The authors have done a good job on the literature review. However, the introduction needs more attention. Please add more literature
related to the essence of this studies - recent articles related to the high performance concrete and high performance fiber reinforced concrete.
Some principal characteristics and innovations of this special concrete should be described. 
More information could be found here: https://doi.org/10.3390/ma11081372, 10.1007/s10853-016-9917-4 - High-performance fiber-reinforced concrete: a review.
3. Please add the scale in case of figures 2 and 3.
4. Concrete composition is not shown in Table 4. This table contain only density of raw materials.
5. Please add detailed content of concrete mixtures.
6. Please provide cement content.
7. Line 172 - please add the unit (mm) to description: '100 x 100 x 100'. 
8. What was the consistency of fresh concrete mixtures?
9. It is recommended to add the grain size curve of aggregates.
10. Please add information about units in case of Figures 6 and 8.
11. Figure 14- please add description of failure modes in the drawings.
12. It is recommended to indicate potential application of research results in engineering.

Reviewer 3 Report

The paper presents tests of facade anchors in fibre-reinforced high-performance concrete. It is of high interest to the community and a current topic of research, and the authors have carried out state-of-art research and unique work. I have however detected some issues, which need to be addressed prior to publication. I am presenting comments below in good faith and with the aim to improve the article and render it to a good condition for publication.

  1. The background literature on anchorage in HPC and FRC is rather poor and sporadic and it should be improved. There are studies a lot more recent and relevant to the paper’s topic, e.g. from Stuttgart, Lulea, BOKU Vienna, Politecnico di Milano, Dortmund, INSA Lyon.
  2. The authors state that the tests were conducted per EAD [33]. However, different standards are available for façade testing. Moreover, EAD specifies the approval procedure but the testing procedure is subject to the respective EOTA technical report. Punching shear is by no means specified in the mentioned EAD (as mentioned in L205) – please revise. Also in the other tensile tests, there are some departures from the EAD/EOTA specifications. Further reference to current relevant standards needs to be made and the deviations thereof in the presented investigations need to be explained and justified by the authors. Some departures from the code: load under angle, specimen and anchor dimensions, fibre reinforcement, high concrete strength
  3. The test configuration should be explained in more detail; particularly (a) what is the support system, and how do the researchers ensure that there are no secondary stresses due to e.g. friction or eccentricities and bending in the system; (b) where are the displacement transducers fixed / is it a steady point?
  4. The 5% quantiles are calculated per EAD with a k factor of 3.4. However, this value corresponds to at least 5 samples. In this study, only 3 samples per parameter were executed, and the k factor is higher, i.e. > 5.0 per the EAD compliant Owen’s statistics to my assessment (see e.g. Odeh & Owen (1980)). This needs to be revised.  
  5. Please clarify how the design resistance value is calculated. The tables read “NRd,c accrording [35]”, but neither [35] is relevant nor further mention is provided. If you meant [34] (EN1992-4), you need to justify how the design equations (e.g. square root of compressive strength)apply for HSC.
  6. Please clarify the quality control system of the tests as regards the fibre distribution and homogeneity of the mix material performance.
  7. Please check the reference list correctness and consistency with a single style. Publication details are missing in the majority of listings, also 35 is from Structures-Elsevier
  8. Several grammar/language issues appear in the document. Please revise. Some examples are below, but odd language is present in the entire document.

L42 Unfortunately, with increasing concrete strength decreases also the material ductility.

L52 The primary reason for the use of fiber in HPC is to increases of the material ductility.

L58 There are two anchoring system for façade elements, which differ from each other by their connection parts

L60 Detail a in Figure 1 present point-supported façade panels….

L80 They performed the mechanical response of the anchor plates…

L123 Due to their chemical composition, the polymer fibers are elastic and flexible in compare to steel fibers…

L179 The anchorage system requireds a combined tensile and shear test….

"accrording " instead of according in many graphs

... and more 

Round 2

Reviewer 1 Report

Dear authors, 

thank you for your answers and considering suggested improvements.

Still, few important comments are missing before accepting the manuscript for the publication.

Please check few more important comments.

1) “… for the use of fiber in HPC is to increases the material ductility and reduce the brittleness [15].”

Please check grammar. It has to be …. to increase….

2) 

We add the explanation: “(see the compressive strength values with their coefficient of variation in Figure 4)”. One of the cubes in ID 3 had small damage on the edges. COV is therefore higher.

Where is the reference in the manuscript? Does that mean you have published already experimtal values? If so, please put reference at the appropriate place.

Still, it is not fully explained what is exactly in Fig.4. Is it scatter band (range from min-max), or as it looks like it is COV whiskers plot type. Would be helpful for reader to know what is exactly presented.

I would like to see listed values from the tests (e.g. add Table with values),such that is more clear the scatter distribution and repeatability. Please also add in the manuscript that you have noticed damage on the edges or other that explains larger scatter.

3) 

Line 259 (first manuscript): The first crack is not the one that is visible on the concrete surface. It is well known from the previous studies that the initial crack start at the anchor head. Therefore, I it not fully correct to write first crack, rather than used first visible crack on the concrete surface. Additionally, it would be very interesting if you could mark the exact position in the Force-Displacement diagram when the first visible crack on the surface was observed. If you want to estimate the moment of the first crack, than at least the linear elastic response should be checked and see the moment of the first deviation from the linearity.

Exactly, the first crack was not analysed visually. It is impossible to mark the force during the test with higher precision. Especially, that the initial crack starts at the anchor head. The first crack was the force that corresponding to the first higher increment of deformation in time increment before the maximum force reached. We changed the x-axis and information in the text.

In the manuscript is added:

  • It is well known from the previous studies, that the initial crack starts at the anchor head. That is fully true in case of headed stud anchor, is that case in your study? Please add references when saying that.
  • The first crack is the moment of the first deviation from the linearity in the linear elastic response.  Can you support that with references or it is the conclusion/statement author want to made? This is not correct statement, as clearly neglecting material plastic deformations. Please carefully revise added statement or remove such statements.

4)

Line 267 (first manuscript): Can you better document how it is proven that the concrete cone did not vary for the different fiber dosages?

We prepare a new graphic in fig. 14 for the reader’s better understanding of the scale.

I do not really understand Fig 14 and saying that the cone size did not vary? In Fig 14 you are showing pull-out resistance, and not cone size. What is the cone size? Is it geometrical size of the failure surface? If so, how we can see that the cone size did not vary from the Load-Displacement curves?

Additionally, in Fig14 I can see difference of almost 50% (Fig. 14a blue curve compared to Fig14.d blue curve!)

Reviewer 2 Report

The majority of remarks have been considered by authors. Errors have been eliminated. 
The authors responded to comments of the reviewer thoroughly.
The current version is more satisfactory for reviewer.

Author Response

We would like to thank the Reviewer for reconsidering our article.

Yours sincerely,

/-/ Szymon Grzesiak

Reviewer 3 Report

Please find attached responses
